# Demonstration-Guided Multi-Objective Reinforcement Learning

**Junlin Lu**      *J.Lu5@universityofgalway.ie*
**Patrick Mannion**      *patrick.mannion@universityofgalway.ie*
**Karl Mason**      *karl.mason@universityofgalway.ie*
*School of Computer Science*
*University of Galway*
*Galway, Ireland, H91 TK33*

**Reviewed on OpenReview:** *https://openreview.net/forum?id=FQAgFgkaFG*

## Abstract

Multi-objective reinforcement learning (MORL) closely mirrors real-world conditions and has consequently gained attention. However, training a MORL policy from scratch is more challenging as it needs to balance multiple objectives according to differing preferences during policy optimization. Demonstrations often embody a wealth of domain knowledge that can improve MORL training efficiency without specific design. We propose an algorithm i.e. demonstration-guided multi-objective reinforcement learning (DG-MORL), which is the first MORL algorithm that can use prior demonstrations to enhance training efficiency seamlessly. Our novel algorithm aligns prior demonstrations with latent preferences via corner weight support. We also propose a *self-evolving mechanism* to gradually refine the demonstration set and avoid sub-optimal demonstration from hindering the training. DG-MORL offers a universal framework that can be utilized for any MORL algorithm. Our empirical studies demonstrate DG-MORL's superiority over state-of-the-art MORL algorithms, establishing its robustness and efficacy. We also provide the sample complexity lower bound and the upper bound of Pareto regret of the algorithm.

## 1 Introduction

Single-objective reinforcement learning (SORL) has exhibited promise in various tasks, such as Atari Games (Mnih et al., 2013), robotics (Schulman et al., 2017), and smart home energy management (Lu et al., 2022a). However, most real-world challenges often encompass multiple objectives. For example, there is always a need balance between safety and time efficiency in driving or between heating-energy usage and cost-saving for home heating. These real-world scenarios extend SORL into multi-objective reinforcement learning (MORL) (Hayes et al., 2022). In MORL, the target is to learn an optimal policy set that maximizes the cumulative discounted reward vector. Users can set preferences to scalarize the reward vector to pick a proper candidate from the optimal policy set.

However, MORL introduces additional complexities and significantly amplifies the challenges already present in SORL. The challenges include: **(i)** *Sparse reward*: In scenarios where rewards are sparse, agents struggle to acquire sufficient information for policy improvement. This challenge, widely recognized in reinforcement learning (RL) (Oh et al., 2018; Ecoffet et al., 2019; 2021; Wang et al., 2023), can slow down the training process, heighten the risk of reward misalignment, and may result in suboptimal policies or even training failure; **(ii)** *Hard beginning*: At the onset of training, policies often face substantial challenges in improving (Li, 2022; Uchendu et al., 2023). This stems from the agent's lack of prior knowledge about the task. Though exploration strategy like $\epsilon$-greedy can be used to gradually mitigate this it is typically inefficient in the early stages of training; **(iii)** *Derailment*: The agent may steer off the path returning to promising areas of state

space due to environmental stochasticity, policy uncertainty, and the randomness inherent in exploration strategies (Ecoffet et al., 2019). This derailment can impede the agent's ability to revisit these advantageous areas, thereby restricting policy improvement.

These challenges are all different reflections of low exploration efficiency. An intuitive solution is to leverage prior demonstrations as guidance to integrate expert knowledge. Unfortunately, existing techniques (Ecoffet et al., 2019; 2021; Uchendu et al., 2023) are exclusively designed for SORL. There is currently no established method for using prior demonstrations in MORL. Moreover, the direct application of demonstration in MORL within the existing paradigm is however unrealistic. Because: **(i)** *Demonstration-Preference Misalignment*: In MORL, a demonstration may be optimal under certain preferences, but detrimental under others. For an agent to effectively learn from a demonstration, it needs to know the preference that renders the demonstration optimal, as a guide of training direction. In most cases, however, even the demonstrators themselves cannot identify the specific numeric preference under which the provided demonstration is the optimal (Lu et al., 2023). Consequently, when an agent is provided with a set of demonstrations without associated preference information, it struggles to correctly match a demonstration with the corresponding preference factor. This can lead the agent in a misguided direction, and potentially fail the training; **(ii)** *Demonstration Deadlock*: In SORL, the demonstration guidance method typically relies on at least one demonstration to assist in learning the optimal policy. However, in MORL, it becomes more complex due to the presence of several, let's say $n$, Pareto non-dominated policies, each optimal under different preferences. If we were to directly apply the existing demonstration-based SORL approach to MORL, it would imply the need for at least $n$ demonstrations to cover these various policies. The challenge is that the exact number $n$ remains unknown until all Pareto non-dominated policies are identified. This creates a paradox: the requirement to have the number of demonstrations $n$ is contingent upon the outcome of the training process itself. It can lead to a lack of necessary prior data, and potentially impair the overall performance. We refer to this paradox as the *demonstration deadlock*; **(iii)** *Sub-optimal Demonstration*: Though demonstrations that are slightly sub-optimal can still effectively guide the training process (Uchendu et al., 2023), it is uncertain whether sub-optimal demonstrations can similarly work in MORL. Sub-optimal demonstrations might obstruct the training, potentially leading to the learning of a sub-optimal policy, or in more severe cases, failure of the training.

In this paper, we propose a novel MORL algorithm, i.e. *demonstration-guided multi-objective reinforcement learning* (DG-MORL) to seamlessly improve MORL training efficiency with demonstrations. DG-MORL is the first MORL algorithm that can use demonstration as guidance. Though a demonstration generally does not correspond to a policy, it can be considered a fixed, hard-coded policy that merely executes actions regardless of the state observation. It is akin to the methods described in (Ecoffet et al., 2019; 2021), where only sequences of actions were utilized to guide the training process. In this study, demonstrations are employed as a *guide policy*. In the following sections demonstrations and *guide policies* are interchangeable. The demonstrations can take any form i.e. trajectory from human experts, a pre-trained policy, or even hand-crafted rules. We assess various types of demonstrations and experimentally show their effectiveness. DG-MORL surpasses the state-of-the-art MORL algorithms in complex tasks. The key contributions of our work include:

**(i)** We propose the **first** MORL algorithm (DG-MORL) that can leverage demonstrations to improve training performance.

**(ii)** We introduce a *self-evolving mechanism* to adapt and improve the quality of guiding data by gradually transferring from prior generated demonstrations to the agent's self-generated demonstrations. This mechanism safeguards the agent against sub-optimal demonstrations.

We outline the related work in Section 2. In Section 3 we present the preliminary of this paper. In Section 4, we formally introduce the DG-MORL algorithm and provide theoretical analysis for our algorithm. We illustrate the baseline algorithms, benchmark environments, experiment setting, metrics, and demonstrate and discuss the experiment results in Section 5. We discuss the limitations of our method in Section 6. The paper is concluded in Section 7.

## 2    Related work

### 2.1    Multi-objective reinforcement learning

MORL methods are categorized as *single-policy* methods and *multi-policy* methods (Yang et al., 2019), distinguished by the number of policies. In single-policy methods, MORL reduces to SORL. The agent uses the SORL learning method to maximize the cumulative reward vector scalarized by a given preference (Mannor & Shimkin, 2001; Tesauro et al., 2007; Van Moffaert et al., 2013). However, when the preference is not given, single-policy algorithms cannot work properly.

The multi-policy methods aim at learning a policy set to approximate the Pareto front. A common method is to train a single policy multiple times upon the different preferences (Roijers et al., 2014; Mossalam et al., 2016). Pirotta et al. (2015) introduced a policy-based algorithm to learn a manifold-based optimal policy. These methods suffer from the curse of dimensionality when the preference space grows and their scalability is limited in complex tasks. To overcome this, policy conditioned on preference to approximate a general optimal policy were proposed, where Abels et al. (2019), and Yang et al. (2019) use vectorized value function updates. Distinguished from the other two methods, Källström & Heintz (2019) use a scalarized value function approximation. By directing the policy training with updating the gradient according to the alignment of value evaluation and preference, these methods are improved. Basaklar et al. (2023) used a cosine similarity between Q-values and preference to achieve an efficient update, while Alegre et al. (2023) used generalized policy improvement (GPI) (Puterman, 2014) to guide the policy to update along with the preference that can bring the largest improvement to the policy.

### 2.2    Prior demonstration utilization approaches

Methods using prior demonstrations for RL usually are a combination of imitation learning (IL) and RL. One direct way to initialize an RL agent with demonstration is to use behavior cloning to copy the behavior pattern of the prior policy (Rajeswaran et al., 2017; Lu et al., 2022a). This method cannot work well with value-based RL frameworks (Uchendu et al., 2023). Furthermore, behavior cloning cannot work with multi-objective settings if the preference is not known beforehand.

Vecerik et al. (2017); Nair et al. (2018); Kumar et al. (2020) have proposed methods for integrating prior data into replay memory. These techniques have been advanced through the use of offline datasets for pre-training, followed by policy fine-tuning (Nair et al., 2020; Lu et al., 2022b). Ball et al. (2023) proposed a simple way to use offline data. However, these methods typically require an offline dataset. In contrast, our approach operates under less stringent assumptions, relying only on a few selected demonstrations.

A Bayesian methodology offers another avenue for employing demonstration data. Ramachandran & Amir (2007) proposed the Bayesian Inverse Reinforcement Learning (BIRL) method, modeling the Inverse Reinforcement Learning (IRL) problem as a Bayesian inference process. By combining prior knowledge with evidence from expert behaviors, they can effectively infer the posterior distribution of the reward function, improving the efficiency and accuracy of policy learning. Choi & Kim (2013) proposed a Bayesian non-parametric approach to construct reward function features in Inverse Reinforcement Learning (IRL). Their method uses the Indian Buffet Process (IBP). It integrates domain expert prior knowledge by having experts provide atomic features representing key factors influencing rewards and enabling more accurate reward reconstruction. Choi & Kim (2012) proposed a non-parametric Bayesian IRL method using the Dirichlet Process Mixture Model (DPMM) to infer an unknown number of reward functions from unlabelled behavior data. This approach can automatically determine the appropriate set of reward functions based on the data, without predefining the number. It is particularly effective in handling expert demonstrations with different goals. They also developed an efficient Metropolis-Hastings sampling algorithm leveraging posterior gradients to estimate the reward functions. However, these works are more from IRL and are not specifically designed for MORL. This is different from our method which uses demonstrations to guide the policy training but not to infer the reward function in a MORL setting.

Furthermore, the absence of extension and empirical validation of these methods in MORL settings hinders direct comparisons to our method. Additionally, multi-objective imitation learning remains under-explored,

presenting another limitation for comparative analysis. Therefore, our algorithm is compared with other existing MORL algorithms.

### 2.3 Effective exploration approaches

Several approaches have been introduced to improve the exploration efficiency, including reward shaping (Ng et al., 1999; Mannion, 2017), curiosity-driven exploration (Barto et al., 2004), and model-based reinforcement learning (MBRL) (Polydoros & Nalpantidis, 2017; Alegre et al., 2023).

However, they still have notable shortcomings. For example, the reliance on domain expertise for reward shaping (Devidze et al., 2022); The inherent vulnerability trapped in local optima in curiosity-driven exploration (Ecoffet et al., 2021); The computation overload and inaccuracy of environment model construction for MBRL. It is necessary to effectively explore the environment so that the model is closer to the ground truth of the environment dynamics. However, this has an additional computational cost when the environment is hard to explore, the environment model may not be accurate enough and this will impact the training adversely.

Using demonstration has the following advantages to mitigate these drawbacks. It allows for the integration of domain knowledge without an explicit reward-shaping function. This can alleviate the reliance on domain expertise; Guided by a demonstration, the agent can return to promising areas that it might have struggled to revisit before the curiosity intrinsic signal is used up; It does not require an environment model and therefore frees from computational demands and model uncertainties.

## 3 Preliminaries

We formalise a *multi-objective Markov decision process* (MOMDP) as $\mathcal{M} := (\mathcal{S}, \mathcal{A}, \mathcal{T}, \gamma, \mu, \boldsymbol{R})$ (Hayes et al., 2022), where $\mathcal{S}$ and $\mathcal{A}$ are state and action space; $\mathcal{T} : S \times A \times S \rightarrow [0,1]$ is a probabilistic transition function; $\gamma \in [0,1)$ is a discount factor; $\mu : S_0 \rightarrow [0,1]$ is the initial state probability distribution; $\boldsymbol{R} : S \times A \times S \rightarrow \mathbb{R}^d$ is a vectorized reward function, providing the reward signal for the multiple ($d \geq 2$) objectives.

The *MORL state-action value function* under policy $\pi$ is defined as: $\boldsymbol{q}^\pi(s,a) := \mathbb{E}_\pi[\sum_{t=0} \gamma^t \boldsymbol{r}_t | s, a]$, where $\boldsymbol{q}^\pi(s,a)$ is a $d$-dimensional vector denoting the expected vectorized return by following policy $\pi$, $\boldsymbol{r}_t$ is the reward vector on $t$ step.

The *multi-objective value vector* of policy $\pi$ with the initial state distribution $\mu$ is: $\boldsymbol{v}^\pi := \mathbb{E}_{s_0 \sim \mu}[\boldsymbol{q}^\pi(s_0, \pi(s_0)]$, where $\boldsymbol{v}^\pi$ is the *value vector* of policy $\pi$, and the $i$-th element of $\boldsymbol{v}^\pi$ is the returned value of the $i$-th objective by following policy $\pi$. The Pareto front (PF) consists of a set of nondominated value vectors. The Pareto dominance relation ($\succ_p$) is: $\boldsymbol{v}^\pi \succ_p \boldsymbol{v}^{\pi'} \iff (\forall i : v_i^\pi \geq v_i^{\pi'}) \cap (\exists i : v_i^\pi > v_i^{\pi'})$. We say that $\boldsymbol{v}^\pi$ is nondominated when all element in $\boldsymbol{v}^\pi$ are not worse than any $\boldsymbol{v}^{\pi'}$ that at least one element of $\boldsymbol{v}^\pi$ is greater than the other $\boldsymbol{v}^{\pi'}$. The Pareto front is defined as $\mathcal{PF} := \{\boldsymbol{v}^\pi | \nexists \pi' \ s.t. \ \boldsymbol{v}^{\pi'} \succ_p \boldsymbol{v}^\pi\}$

To select an optimal policy from the PF, the criteria are usually given by the user and therefore termed as *user-defined preference*. A *utility function* is used (Hayes et al., 2022) to map the value vector to a scalar. One of the most frequently used utility functions is the linear utility function (Hayes et al., 2022), i.e. $u(\boldsymbol{v}^\pi, \boldsymbol{w}) = v_{\boldsymbol{w}}^\pi = \boldsymbol{v}^\pi \cdot \boldsymbol{w}$, where $\boldsymbol{w}$ is on a simplex $\mathcal{W} : \sum_i^d w_i = 1, w_i \geq 0$.

The *Convex Coverage Set (CCS)* is a finite convex subset of the PF (Hayes et al., 2022). $CCS := \{\boldsymbol{v}^\pi \in \mathcal{PF} | \exists \boldsymbol{w} \ s.t. \ \forall \boldsymbol{v}^{\pi'} \in \mathcal{PF}, \boldsymbol{v}^\pi \cdot \boldsymbol{w} \geq \boldsymbol{v}^{\pi'} \cdot \boldsymbol{w}\}$. When a linear utility function is used, the CCS comprises all non-dominated values, making the PF and CCS equivalent.

In the cutting-edge MORL methods (Yang et al., 2019; Alegre et al., 2023), $\pi$ is typically trained through learning a vectorized Q function that can adapt to any preference weight vector. These methods still suffer from poor sample efficiency as they search for the CCS by starting from the original point in solution space. See Figure 1(a), traditional MORL method starts from the red point, and searches to expand to the orange CCS. Ideally, by giving demonstrations, the training process can be transformed into Figure 1(b). The policy is informed by domain knowledge, allowing it to start from the red points (demonstrations). This shift significantly reduces the distance between the "known" solution and the CCS, thereby reducing sample

complexity of the training process. Another limitation is the demonstrations' sub-optimality. This introduces the risk of the policy being confined within the bounds of sub-optimal demonstrations (1 (c), where the green dot represents the policy approximated by a Q network). The gap between the red starting points and the CCS can be challenging to bridge if the policy solely satisfies about meeting the initial demonstration standard. As the policy may chance upon better trajectories, when the policy turns to this better guidance, it will exceed the initial demonstrations. Figure 1(d) illustrates this, where the dot alternating between red and green represents a demonstration identified by the policy. Importantly, these new demonstrations are discovered by the agent itself. The initial demonstrations serve merely as a catalyst. This concept is formalized as the *self-evolving mechanism.*

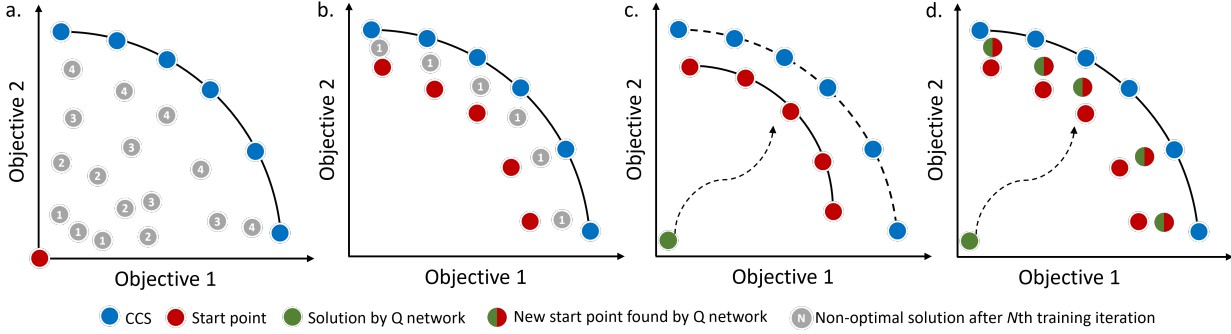

Figure 1: (a) Training process of traditional MORL; (b) Training process start from demonstration; (c) DG-MORL without self-evolving; (d) DG-MORL with self-evolving.

# 4 Demonstration-guided multi-objective reinforcement learning

In this section, we present the DG-MORL algorithm. We begin by addressing the demonstration-preference misalignment, followed by introducing the *self-evolving mechanism* and then we address demonstration deadlock and the sub-optimal demonstration problem. Finally, we provide an in-depth description of the DG-MORL algorithm, accompanied by theoretical analysis.

## 4.1 Demonstrations corner weight computation

When addressing the issue of *Demonstration-Preference Misalignment*, it is noteworthy that a single demonstration can be optimal under an infinite number of preference weights as the weight simplex is continuous. Sampling and iterating through the entire simplex of preferences is computationally expensive and impractical. To mitigate this challenge, we utilize the concept of *corner weights* (Roijers, 2016) i.e. denoted as $\mathcal{W}_{corner}$, to discretize the continuous simplex into several critical points where the policy undergoes essential changes.

We now give a clear explanation of the corner weights. When using different weights to scalarize the return of the objectives, we get different scalar values. If we plot these scalar values against the weights, we get a piecewise linear convex curve or surface. Corner weights are specific combinations of weights where the slope of this curve changes. In simpler terms, they are the "corners" or "edges" on the graph of the scalarized value function. As shown in Figure 2, the points depicted by the blue lines are the corner weights according to the curve, the y-axis is the return value while the x axis is the weight of objective 1 (assuming this is a 2-objective problem). These points are significant because they represent weights where the optimal strategy or policy change. According to Roijers (2016), that the maximum possible improvement to the set of known returns occurs at one of the corner weights. This means that if we're going to find a better policy, it's most likely to happen when we look at these specific weights. See a more rigorous definition of the corner weights in Definition 1.

**Definition 1.** *(Definition 19 of Roijers (2016))*

*Given a set of value vectors $\mathcal{V}$, we can define a polyhedron $P$ by:*

$$P = \{\boldsymbol{x} \in \mathbb{R}^{d+1} | \boldsymbol{V}^+\boldsymbol{x} \leq \boldsymbol{0}, \sum_i w_i = 1, w_i \geq 0, \forall i\} \tag{1}$$

*where $\boldsymbol{V}^+$ represents a matrix in which the elements of the set $\mathcal{V}$ serve as the row vectors augmented by -1's as the column vector. The element $\boldsymbol{x}=(w_1, ..., w_d, v_{\boldsymbol{w}})$ in $P$ consists of the element of the weight vector and its scalar return value. Corner weights refer to the weights located at the vertices of $P$.*

By iterating conduct the demonstrations, we can get a set of vectorized returns $\mathcal{V}_d$. We denote the convex coverage set of the demonstration results as $CCS_d$. The corner weights effectively partition the solution space and classify the policies. The choice of the optimal policy maximizing utility for a given weight varies when traversing across corner weights. By computing $\mathcal{W}_{corner}$ on $CCS_d$, we can assign corresponding weights to the demonstrations. Employing corner weights efficiently reduces the computational burden by calculating on a finite set $\mathcal{W}_{corner}$ rather than on an infinite weight simplex $\mathcal{W}$. Additionally, the significance of using corner weights in training is emphasized by the theorem:

**Theorem 1.** *(Theorem 7 of Roijers (Roijers, 2016)) There is a corner weight $\boldsymbol{w}$ that maximizes:*

$$\Delta(\boldsymbol{w}) = u_{\boldsymbol{w}}^{CCS} - \max_{\pi \in \Pi} u_{\boldsymbol{w}}^{\pi} \tag{2}$$

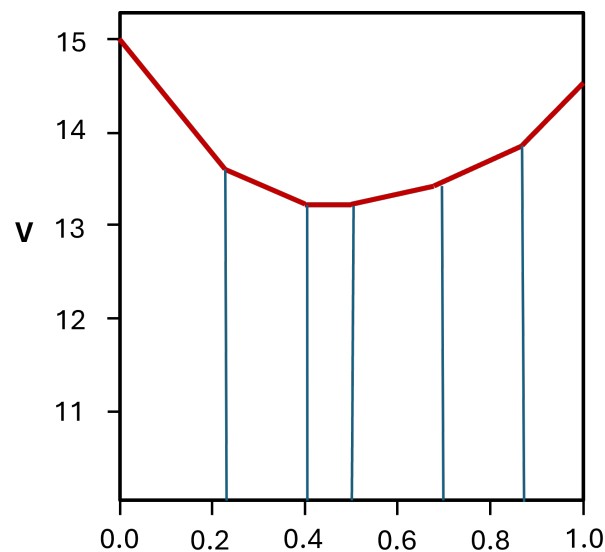

where $\Delta(\boldsymbol{w})$ is the difference between the utility of optimal policy given weight $\boldsymbol{w}$ and the utility obtained by a policy $\pi \in \Pi$, where $\Pi$ denotes the current agent's policy set. Theorem 1 suggests that training along a corner weight allows for faster learning of the optimal policy (set), as it offers the potential for achieving maximum improvement. Assuming the prior demonstration is the known optimal solution, by replacing the $CCS$ in equation 2 with $CCS_d$, Equation 2 can be rewritten as: $\Delta_d(\boldsymbol{w}) = u_{\boldsymbol{w}}^{CCS_d} - \max_{\pi \in \Pi} u_{\boldsymbol{w}}^{\pi}$. $\boldsymbol{w}_c = \underset{\boldsymbol{w} \in \mathcal{W}_{corner}}{\arg\max} \Delta(\boldsymbol{w})$. It further exploits the theorem to select the corner weight $\boldsymbol{w}_c$ from $\mathcal{W}_{corner}$ that denotes the largest gap between the performance of the current policy and the demonstration to increase the training efficiency.

Figure 2: Corner Weights Showcase: The points depicted by the blue lines are the corner weights according to the curve, the y-axis is the return value while the x axis is the weight of objective 1 (assuming this is a 2-objective problem).

We have elucidated the calculation of the corner weights set and the selection of an appropriate candidate corner weight for aligning the demonstration. With this approach, the issue of *demonstration-preference misalignment* is addressed.

## 4.2 Self-evolving mechanism

An inaccurate assumption exists that "the use of demonstrations should naturally lead to performance improvements". It is too optimistic as sub-optimal demonstrations can easily impact the training process and cause low performance. We propose a *self-evolving mechanism* to tackle the problem of sub-optimal initial demonstrations. It is designed to improve upon given demonstrations for better guidance. This is another main contribution of our work. In the following sections, the demonstrations are referred to as *guide policies*. Although a demonstration generally does not correspond to a policy, it can be considered a fixed, hard-coded policy that merely executes actions regardless of the state observation. The policies developed by the agent are called *exploration policies*.

We use the notation $\Pi_g$ to represent the set of *guide policies*. Since the *exploration policy* set is approximated by a single neural network approximator, we denote it as $\Pi_e = \{\pi_e^{\boldsymbol{w}_i} | \forall \boldsymbol{w}_i \in \mathcal{W}\}$. It is conditioned on the input preference weight vector and serves as the policy to be improved via learning. The agent interacts with the environment during training with a mix of *guide policy* and *exploration policy*, we denote this mixed policy as $\pi$.

The *self-evolving mechanism* is designed to continuously update and improve the *guide policy* set $\Pi_g$. When the agent utilizes the mixed policy $\pi$, it may discover better demonstrations. When new, superior demonstrations are found, they are dynamically incorporated into the repository and treated as a new curriculum or guide to the agent. This enables more effective learning from these superior demonstrations rather than persisting with initial data. The newly discovered demonstrations are closer to the optimal policy. Even when starting with sub-optimal demonstrations, as long as they guide the agent to valuable regions within the state space, the agent can begin exploring from there and identify better solutions. This progression makes the *guide policy* gradually move closer to the optimal policy, thereby mitigating the impact of sub-optimal data. Moreover, the agent might uncover demonstrations that were not initially provided. These new demonstrations can introduce a fresh set of corner weights, enriching the guiding curriculum and addressing the *demonstration deadlock*. As learning progresses, less effective demonstrations are phased out, allowing the demonstration set to evolve. We symbolize this process as $\Pi_g \to \Pi_g'$. This process, while it involves updating the learning policy based on new demonstrations, does not strictly align with the traditional definitions of on-policy or off-policy training.

### 4.3 Multi-stage curriculum

To make full use of the demonstration, the mixed policy $\pi$ is mostly controlled by the *guide policy* $\pi_g$ at the beginning of training. This can navigate the agent to a more promising state than randomly exploring. The control of the *guide policy* is diluted during training and taken over by the *exploration policy*. However, swapping the *guide policy* with the *exploration policy* introduces the challenge of the policy shift. A multi-stage curriculum pattern, as suggested by Uchendu et al. (2023), is implemented. This strategy involves gradually increasing the number of steps governed by the *exploration policy*, facilitating a smoother transition and mitigating the policy shift issue.

### 4.4 Algorithm

Compared with JSRL Uchendu et al. (2023), DG-MORL introduces the integration of the corner weight support to align the provided demonstration with proper preferences; The *self-evolving mechanism* of DG-MORL substantially reduces the need for numerous demonstrations to attain strong performance[1]. Another departure is that DG-MORL entails a controlling swap between the set of *guide policies* to one *exploration policy*. Given a preference weight vector $\boldsymbol{w}$, a *guide policy* upon which the exploration may gain the largest improvement is selected as shown in Equation 3.

$$\pi_g = \arg\max_{\pi \in \Pi_g} \Delta^{\pi}(\boldsymbol{w}) = \arg\max_{\pi \in \Pi_g} (u_{\boldsymbol{w}}^{\pi} - u_{\boldsymbol{w}}^{\pi_e}) \tag{3}$$

We provide a detailed explanation of the DG-MORL in Algorithm 1. To initiate the process, a set of initial demonstrations is evaluated, and the return values are added to $CCS_d$. Dominated values are subsequently discarded, and assume that these retained values represent "optimal" solutions. The corner weights are then computed based on these values[2].

Next, a suitable *guide policy* is selected by using Equation 3. The episode horizon is denoted as $H$. The *guide policy* control scale is denoted as $h$, i.e. for the first $h$ steps, the *guide policy* controls the agent, while for the next $H - h$ steps, the *exploration policy* is performed. The *exploration policy*'s performance is periodically evaluated. When improved performance is demonstrated, the parameter $h$ is reduced to grant the *exploration policy* more controllable steps. The superior demonstration is then added to the demonstration repository,

---

[1]JSRL necessitates a large number of e.g. 2000 demonstrations.

[2]We utilize the implementation from Alegre et al. (2023), which relies on the pycddlib library via https://github.com/mcmtroffaes/pycddlib/ to identify the vertices of polyhedron P.

enabling the demonstration set to evolve autonomously (self-evolving). The $CCS$ is periodically updated to eliminate dominated solutions. A new set of corner weights is computed based on the updated $CCS$.

In certain instances, the *guide policy* may be so advanced that it becomes challenging for the *exploration policy* to exceed its performance. This scenario can restrict the rollback steps and therefore limit the *exploration policy*'s touching scope of states, consequently impeding the learning efficiency. For example, the agent could find itself unable to progress beyond the initial stages of learning a solution due to the high difficulty level of the *guide policy*. To address this issue, we introduce the concept of a *passing percentage*, denoted as $\beta \in [0, 1]$. This strategy involves setting a comparatively lower performance threshold initially, making it more achievable for the agent to learn. This threshold is incrementally raised, thereby progressively enhancing the agent's performance.

---

**Algorithm 1** DG-MORL Algorithm

---

1: **Input:** prior demonstration $\Pi_g$, pass threshold $\beta$
2: Evaluate the $\Pi_g$, add demonstration results to $CCS_d$
3: $CCS \leftarrow CCS_d$
4: **while** max training step not reached **do**
5:     Get corner weight set $\mathcal{W}_{corner}$ based on Equation 1 with $CCS$
6:     Given $\mathcal{W}_{corner}, \Pi_g$, get candidate weight $\boldsymbol{w}_c$ based on Equation 2
7:     With $\boldsymbol{w}_c$ and $\Pi_g$, sample the *guide policy* $\pi_g$ based on Equation 3
8:     Calculate the utility threshold: $u_\theta = \boldsymbol{v}_{\pi_g} \cdot \boldsymbol{w}_c$
9:     The *guide policy* $\pi_g$ controllable scale $h = H$, i.e. $\pi_g$ can control the whole episode.
10:     **while** $h \geq 0$ **do**
11:        Set mix policy $\pi = \pi_g^{[0:h]} + \pi_e^{[h+1:H]}$
12:        Roll out $\pi$, gather the experience
13:        Train explore policy $\pi_e$ [a]
14:        Evaluate $\pi$: $u = \boldsymbol{v}_\pi \cdot \boldsymbol{w}_c$
15:        **if** $u > u_\theta \cdot \beta$ **then**
16:           Add the better value vector $\boldsymbol{v}_\pi$ to $CCS$
17:           $h \leftarrow h - \Delta h$, $\Delta h$ is the rollback span
18:        **end if**
19:     **end while**
20:     Remove dominated solutions, update $CCS$ and $\Pi_g$
21:     Update the curriculum pass threshold $\beta$
22: **end while**
  [a] $\pi_e$ can be trained with any MORL algorithm, we use GPI-LS in this work.

---

## 4.5 Theoretical analysis

We provide a theoretical analysis of the lower bound and upper bound on the sample complexity of the algorithm

### 4.5.1 Lower bound discussion

When the *exploration policy* is 0-initialized (all Q-values are zero initially), a non-optimism-based method, e.g. $\epsilon$-greedy, suffers from exponential sample complexity when the horizon expands. We extend the combination lock (Koenig & Simmons, 1993; Uchendu et al., 2023) under the multi-objective setting (2 objectives), and visualize an example in Figure 3.

In the multi-objective combination lock example, the agent encounters three distinct scenarios where it can earn a non-zero reward vector, each scenario is determined by a specific preference weight vector. The three preference weight vectors are $[1, 0]$, $[0.5, 0.5]$, and $[0, 1]$. These vectors correspond to the agent's varying preferences towards two objectives: When the agent demonstrates an extreme preference for objective 1, it may receive a reward vector of $[1, 0]$. Conversely, if the agent shows an extreme preference for objective 2, it could be awarded a reward vector of $[0, 1]$.

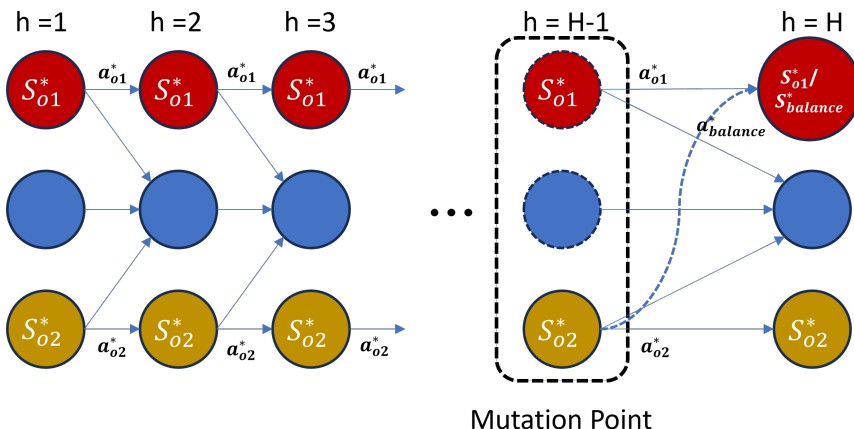

Figure 3: Lower bound example: MO combination lock

In the scenario where the agent's preference is evenly split between the two objectives, as indicated by a preference vector of $[0.5, 0.5]$, there's a possibility for it to obtain a reward vector of $[0.5, 0.5]$. This outcome reflects a balanced interest in both objectives. In this multi-objective combination lock scenario, the outcomes for the agent vary significantly based on its preference weights and corresponding actions. When the preference weights are set to $[1, 0]$, the agent's goal is to reach the red state $s^*_{o1:h}$. To achieve this, it must consistently perform the action $a^*_{o1}$ from the beginning of the episode, which enables it to remain in the red state and subsequently receive a reward feedback of $[1, 0]$. Similarly, with preference weights of $[0, 1]$, the agent needs to execute the action $a^*_{o2}$ right from the start to eventually reach a reward vector of $[0, 1]$. For a balanced preference of $[0.5, 0.5]$, the agent is expected to take a specific action, $a^*_{balance}$, at step $H - 1$ to transition to the state $s^*_{balance:h}$ and receive a balanced reward of $[0.5, 0.5]$. This time point is referred to as the mutation point. If the agent derails from these prescribed trajectories, it only receives a reward of $[0, 0]$. Moreover, any improper action leads the agent to a blue state, from which it cannot return to the correct trajectory (derailment).

As the agent is 0-initialized, it needs to select an action from a uniform distribution. The preference given is also sampled from a uniform distribution. Then if the agent wants to know the optimal result for any of the three preferences, it needs at least $3^{H-1}$ samples. If it thoroughly knows the optimal solution of one of the three preferences, it can achieve an expected utility of 0.3. There exist several ($K$) paths in this MOMDP instance delivering good rewards. To find all good policies, the agent needs at least to see $K \cdot 3^{H-1}$ samples. We formalize this in Theorem 2.

The scenario where preferences precisely match the three specific weights of $[1, 0]$, $[0.5, 0.5]$, and $[0, 1]$ is considerably rare, as preferences are typically drawn from an infinite weight simplex. In practical training situations, encountering a variety of preferences can lead to conflicting scenarios for the agent. This conflict arises because different preferences may necessitate different actions or strategies, potentially causing the agent's policy to experience "forgetfulness" or an inability to consistently apply learned behaviors across varying preferences. Such a situation complicates the training process and increases sample complexity.

**Theorem 2.** *(Koenig & Simmons, 1993; Uchendu et al., 2023) When using a 0-initialized exploration policy, there exists a MOMDP instance where the sample complexity is exponential to the horizon to see a partially optimal solution and a multiplicative exponential complexity to see an optimal solution.*

### 4.5.2 Upper bound of DG-MORL

We now give the upper bound of DG-MORL's regret. We start by making assumptions about the quality of the *guide policy*, i.e., it can cover some of the features that can be visited under the optimal policy, and the coverage rate is restricted by a concentrate ability coefficient. Then we give a Lemma to show that

in a sequential decision making setting, the difference between the optimal policy and the *guide policy* is bounded and this difference will not increase under the self-evolution mechanism; but may decrease. Next, we provide several key definitions including Pareto suboptimality gap and sequence Pareto regret. Then we give a performance guarantee.

**Assumption 1.** *(Uchendu et al., 2023) Assume there exists a feature mapping function $\phi : \mathcal{S} \to \mathbb{R}^d$, that for any policy $\pi$, $Q^\pi(s,a)$ and $\pi(s)$ depends on $s$ only through $\phi(s)$. The guide policy $\pi_g$[3] is assumed to cover the states visited by the optimal policy corresponding to the preference weight vector $w$:*

$$\sup_{s,h,w} \frac{d_{h,w}^{\pi^*}(\phi(s))}{d_{h,w}^{\pi_g}(\phi(s))} \le C \tag{4}$$

$d_{h,w}^{\pi^*}(\phi(s))$ and $d_{h,w}^{\pi_g}(\phi(s))$ represents the probability distribution of feature $\phi(s)$ being visited at time step $h$ under the preference weight $w$, while following the optimal policy $\pi^*$ or *guide policy* $\pi_g$ respectively. $C$ is the concentratability coefficient (Rashidinejad et al., 2021; Guo et al., 2023). It quantifies the extent to which the *guide policy* $\pi_g$ covers the optimal policy $\pi^*$ in the feature space and restricts the probability of the optimal policy visiting a certain feature $\phi(s)$ to not exceed $C$ times the probability of the *guide policy* visiting that feature.

Nonetheless, Assumption 1 is formulated within contextual bandit which does not sufficiently capture the complexities of sequential decision-making processes. We extend Assumption 1 to the context of sequential decision-making and incorporates a *self-evolving mechanism*, thereby establishing Lemma 3. As a result of the *self-evolving mechanism*, the *guide policy* $\pi_g$ progressively enhances its performance over time.

**Lemma 3.** *For a MOMDP, the sequence concentration coefficient $C_{seq}(t)$ under preference weight $w$ is defined as the maximum value of the concentration coefficient across all time steps $h$ while with the self-evolving mechanism:*

$$\max_{h \in \{1,2,...,H\}} \frac{d_{h,w}^{\pi^*}(\phi(s))}{d_{h,w}^{\pi_g}(\phi(s))} \le C_{seq}(t,w) \le C_{seq}(t_0,w) \tag{5}$$

*Furthermore, we can give the Pareto concentration coefficient $C_{Pareto}(t) = \max_{w \in \mathcal{W}} C_{seq}(t,w) \le C_{Pareto}(t_0)$*

*So we have:*

$$\max_{h \in \{1,2,...,H\}} \frac{d_h^{\pi^*}(\phi(s))}{d_h^{\pi_g}(\phi(s))} \le C_{Pareto}(t) \le C_{Pareto}(t_0) \tag{6}$$

We now give a few key definitions to calculate the upper bound of regret. One critical concept is the *Pareto suboptimality gap*. This term refers to the measure of distance between the current policy's performance on a specific arm of the multi-objective contextual bandit (MOCB) and the true Pareto optimal solution for that arm.

**Definition 2.** *(Pareto suboptimality gap (Lu et al., 2019)) Let $x$ be an arm in $\mathcal{X}$, i.e. the arm set of the MOCB. The Pareto suboptimality gap $\Delta x$ is defined as the minimal scalar $\sigma \ge 0$ so that $x$ becomes Pareto optimal by adding $\sigma$ to all entries of its expected reward.*

We now extend the Definition 2 to MOMDP to fit in sequence decision making process:

**Definition 3.** *(Sequence Pareto Regret) In a MOMDP, given a trajectory $\tau = (s_1, a_1, s_2, a_2, ..., s_H)$, the expectation of sequence Pareto regret $\Delta_{seq}^\pi$ for a finite horizon is:*

$$\Delta_{seq}^\pi = \Delta_{s_1}^\pi + \mathbb{E}[\Delta_{s_2}^\pi \mid s_1, a_1] + \mathbb{E}[\Delta_{s_3}^\pi \mid s_2, a_2] + \ldots + \mathbb{E}[\Delta_{s_H}^\pi \mid s_{H-1}, a_{H-1}] \tag{7}$$

*where $\Delta_{s_h}$ is the Pareto suboptimiliy gap in state $s_h$, defined as follows:*

$$\Delta_{s_h}^\pi = \min \left\{ \sigma \ge 0 \,\middle|\, \forall a \in A, \, \nexists a' \in A, \, \boldsymbol{Q}_{s_h,a}^* \ge \boldsymbol{Q}_\pi(s_h, a) + \sigma \cdot \boldsymbol{1} \,, \, \boldsymbol{Q}_{s_h,a}^* \ne \boldsymbol{Q}_\pi(s_h, a) + \sigma \cdot \boldsymbol{1}, \, \right\} \tag{8}$$

---

[3]Given that the policy set $\Pi_g$ can be interpreted as a mixed policy tailored to specific scenarios, it retains its generality even when $\Pi_g$ is considered as a singular policy.

In Equation 7 , $\Delta_{\text{seq}}^{\pi}$ is the expected sequence Pareto regret of policy $\pi$. It quantifies the cumulative performance loss of policy $\pi$ relative to the Pareto optimal policy throughout the decision-making process. $\Delta_{s_1}^{\pi}$ is the Pareto regret at the initial state $s_1$. $\mathbb{E}[\Delta_{s_h}^{\pi} \mid s_{h-1}, a_{h-1}]$ is the expected Pareto regret at state $s_h$ which is conditioned on the previous state and action. By summing the initial Pareto regret and the conditional expected regrets at each subsequent state, we obtain the total expected sequence Pareto regret.

Equation 8 clarifies the conditions for Pareto dominance. $\boldsymbol{Q}_{s_h,a}^*$ represents the multi-objective Q-value of the optimal policy $\pi^*$ for action $a$ in state $s_h$. $\boldsymbol{Q}_{\pi}(s_h, a)$ is the multi-objective Q-value of policy $\pi$ for action in state $s_h$. $\sigma$ indicates the minimum amount that needs to be added to all objectives for the current policy $\pi$ to reach the level of the optimal policy across all actions while $\boldsymbol{1}$ is a full ones vector whose dimensionality is the same as the reward space. For each action $a \in A$ of strategy $\pi$, there does not exist an action $a' \in A$ such that $\boldsymbol{Q}_{s_h,a}^*$ is no worse than $\boldsymbol{Q}_{\pi}(s_h, a) + \sigma \cdot \boldsymbol{1}$ across all objectives, and strictly better on at least one objective. We accurately measure the Pareto suboptimality when following $\pi$ relative to the optimal policy at state $s_h$.

**Theorem 4.** *The expectation of the sequence Pareto regret $\Delta_{seq}^{\pi}$ is the sum of the expectation of the Pareto regret for each step.*

$$\mathbb{E}[\Delta_{seq}^{\pi}] = \sum_{h=1}^{H} \mathbb{E}[\Delta_{s_h}^{\pi}] \tag{9}$$

*Equation 9 indicates the additivity of regret, i.e. in MOMDP, the expected Pareto regret of a policy over the entire decision sequence can be decomposed into the sum of the expected Pareto regret at each time step.*

We put the proof of Theorem 4 below.

*Proof.* Take expectations on both side:

$$\mathbb{E}[\Delta_{\text{seq}}^{\pi}] = \mathbb{E}[\Delta_{s_1}^{\pi}] + \mathbb{E}[\mathbb{E}[\Delta_{s_2}^{\pi} \mid s_1, a_1]] + \mathbb{E}[\mathbb{E}[\Delta_{s_3}^{\pi} \mid s_2, a_2]] + \ldots + \mathbb{E}[\mathbb{E}[\Delta_{s_H}^{\pi} \mid s_{H-1}, a_{H-1}]]$$

According to the Law of total expectation:

$$\mathbb{E}[\Delta_{\text{seq}}^{\pi}] = \mathbb{E}[\Delta_{s_1}^{\pi}] + \mathbb{E}[\Delta_{s_2}^{\pi}] + \mathbb{E}[\Delta_{s_3}^{\pi}] + \ldots + \mathbb{E}[\Delta_{s_H}^{\pi}] = \sum_{h=1}^{H} \mathbb{E}[\Delta_{s_h^{\pi}}]$$

$\square$

We can now give the upper bound of the sum of the sequence Pareto regret during the training for $T$ rounds.

**Theorem 5.** *(Performance Guarantee) Assuming that the environment satisfies the Markov property, and there exists a feature mapping function $\phi$ such that for any policy $\pi$ both $\boldsymbol{Q}_{\pi}(s, a)$ and $\pi(s)$ depend only on $\phi(s)$. The DG-MORL algorithm guarantees that the guiding policy $\pi_g$ progressively approaches the optimal policy $\pi^*$ through self-evolution mechanisms. $f(t)$ is an abstract regret function, it depends on the specific algorithm we used to train $\pi_e$. During each training round $t \in [T]$, the algorithm executes policy $\pi_t$, and the sum of sequence Pareto regret $PR_{seq}(T)$ is bounded by:*

$$PR_{seq}(T) = \sum_{t=1}^{T} \Delta_{seq}^{\pi_t} \leq T H R_{\max} C \tag{10}$$

*where $C = C_{Pareto}(t_0) + f(t_0)$, $C_{Pareto}(t_0)$ and $f(t_0)$ are the Pareto concentration coefficient and regret function at the first round of training. This is to get a conservative upper bound as $C_{Pareto}(t_0) \geq C_{Pareto}(t)$ because of the self-evolving mechanism and $f(t_0) \geq f(t)$ because of the policy improvement. $R_{\max} = \max_{s,a} \|r(s, a)\|_1$, we use this 1-norm to get a relative conservative and general upper bound[4]*

---

[4]While we acknowledge that using preference weight might add specificity, we prefer to use the 1-norm since it relaxes the upper bound. This approach ensures that our theorem remains valid even in the worst-case scenario.

*Proof.* of Theorem 5

Decomposing the regret into contributions from the *guide policy* $\pi_g$ and the *exploration policy* $\pi_e$. In the $t$-th round of training, the Pareto regret of the sequence of the mixed strategies $\pi_t$ can be divided into two parts:

(1). Regret from $\pi_{g,t}$

$$\Delta_{seq}^{\pi_{g,t}} = \sum_{h=1}^{h_g} \Delta_{s_h}^{\pi_{g,t}}$$

where $h_g$ is the last time step controlled by $\pi_{g,t}$
(2). Regret from $\pi_{e,t}$

$$\Delta_{seq}^{\pi_{e,t}} = \sum_{h=h_g+1}^{H} \Delta_{s_h}^{\pi_{e,t}}$$

Thus, the total regret is :

$$\Delta_{seq}^{\pi_t} = \Delta_{seq}^{\pi_{g,t}} + \Delta_{seq}^{\pi_{e,t}}$$

With the help of the *self-evolving mechanism*, $\pi_g$ can improve overtime. According to Lemma 3, at $h$ timestep, the expected regret of $\pi_{g,t}$ is:

$$\mathbb{E}_{s_h}[\Delta_{s_h}^{\pi_{g,t}}] \leq R_{\max} \cdot C_{Pareto}(t)$$

In the worst case, the reward difference is $R_{\max}$, and the difference in state distribution is controlled by $C_{Pareto}(t)$. In the worst case, each time step can cause the maximized regret. The total expected regret for $\pi_{g,t}$ is:

$$\Delta_h^{\pi_{g,t}} \leq h \cdot R_{\max} \cdot C_{Pareto}(t)$$

Assuming we use $\epsilon$-greedy strategy in the *exploration policy* $\pi_e$. For the *exploration policy* $\pi_e$, at each time step $h$, the expected regret of $\pi_{e,t}$ is:

$$\mathbb{E}_{s_h}[\Delta_{s_h}^{\pi_{e,t}}] \leq f(t) \cdot R_{\max}$$

This is because a random action is selected with probability $\epsilon$, which may lead to a reward loss of up to $R_{\max}$ in the worst case. Therefore, the total expected regret for the *exploration policy* is:

$$\Delta_{H-h}^{\pi_{e,t}} \leq (H - h)f(t) \cdot R_{\max}$$

For $T$ training rounds, the total regret is:

$$PR_{seq}(T) = \sum_{t=1}^{T} \Delta_{seq}^{\pi_t} \leq \sum_{t=1}^{T} \left( h_t \cdot R_{\max} \cdot C_{Pareto}(t) + (H - h_t) \cdot f(t) \cdot R_{\max} \right)$$

where $h_t$ is the number of time steps controlled by the guided policy during the $t$-th training round. During the training process, the number of time steps controlled by the guided policy decreases over time, i.e., $h_t$ decreases with $t$. Since $C_{Pareto}(t)$ and $f(t)$ decreases with $t$, we approximate it by $C_{Pareto}(t_0)$ and $f(t_0)$, the value at the final round. We approximate that the steps controlled by $\pi_g$ and $\pi_e$ are all the maximized number of steps, i.e. $H$, the upper bound on the total regret is:

$$PR_{seq}(T) \leq THR_{\max}C$$

,where $C = C_{Pareto}(t_0) + f(t_0)$ □

## 5 Experiments

In this section, we introduce the baselines, benchmark environments and metrics. We then illustrate and discuss the results. We have also provided the sensitivity analysis about the how different qualities and quantities of the initial demonstrations can influence the training process in the Appendix. Please note that, our DG-MORL does not require specific preferences to be covered by demonstrations. In MORL, as some preferences may share the same optimal policy, if the true PF is not known, it is impossible to cover all important preferences by demonstrations.

### 5.1 Baseline algorithms

We use two state-of-the-art MORL algorithms as baselines: GPI linear support (GPI-LS) and GPI prioritized Dyna (GPI-PD) (Alegre et al., 2023). GPI-PD algorithm is model-based and GPI-LS is model-free.

The GPI-LS and GPI-PD are based on generalized policy improvement (GPI) (Puterman, 2014) while the GPI-PD algorithm is model-based and GPI-LS is model-free. The GPI method combines a set of policies as an assembled policy where an overall improvement is implemented (Barreto et al., 2017; 2020). It was introduced to the MORL domain by Alegre et al. (2022). The GPI policy in MORL is defined as $\pi_{GPI}(s|\boldsymbol{w}) \in \arg\max_{a \in \mathcal{A}} \max_{\pi \in \Pi} q_{\boldsymbol{w}}^{\pi}(s, a)$. The GPI policy is at least as well as any policy in $\Pi$ (policy set). The GPI agent can identify the most potential preference to follow at each moment rather than randomly pick a preference as in Envelope MOQ-learning. This facilitates faster policy improvement in MORL. Furthermore, the model-based GPI-PD can find out which piece of experience is the most relevant to the particular candidate preference to achieve even faster learning. The two baseline algorithms GPI-LS and GPI-PD cannot utilize demonstrations. There is not yet a MORL algorithm that can use demonstrations in training and DG-MORL is the first one.

As indicated in the literature (Alegre et al., 2023), the Envelope MOQ-learning algorithm (Yang et al., 2019), SFOLS (Alegre et al., 2022), and PGMORL (Wurman et al., 2022) are consistently outperformed by the GPI-PD/LS. Therefore, if our algorithm can surpass GPI-PD/LS, it can be inferred that it also exceeds the performance of these algorithms. We also use a no-self-evolving version of DG-MORL in the ablation study as a baseline.

#### 5.1.1 Implementation detail

The code of this work is available on https://github.com/MORL12345/DG-MORL.git. We use the same neural network architecture as the GPI-PD implementation in all benchmarks, i.e. 4 layers of 256 neurons in DST and Minecart, 2 layers with 256 neurons for both the critic and actor in MO-Hopper, MO-Ant and MO-Humanoid. We use Adam optimizer, the learning rate is $3 \cdot 10^{-4}$ and the batch size is 128 in all implementations. As for the exploration, we adopted the same annealing pattern $\epsilon$-greedy strategy. In DST, the $\epsilon$ anneals from 1 to 0 during the first 50000 time steps. In Minecart, the $\epsilon$ is annealed from 1 to 0.05 in the first 50000 time steps. For the TD3 algorithm doing MO-Hopper, MO-Ant and MO-Humanoid, we take a zero-mean Gaussian noise with a standard deviation of 0.02 to actions from the actor-network. All hyper-parameters are consistent with literature (Alegre et al., 2023).

The initial demonstration data are from different sources to validate the fact that any form of demonstration can be used as it can be converted to action sequences. The demonstration for DST is from hard-coded action sequences which are sub-optimal. Minecart uses the demonstrations manually played by the author, i.e. us, while the MO-Hopper, MO-Ant and MO-Humanoid, use the demonstration from a underdeveloped policy from the TD3 algorithm.

### 5.2 Benchmark environments and metrics

We conduct the evaluation on MORL tasks with escalating complexity: from an instance featuring discrete states and actions, i.e. *Deep Sea Treasure (DST)* (Yang et al., 2019; Alegre et al., 2023) to tasks with continuous states and discrete actions, i.e. *Minecart* (Abels et al., 2019; Alegre et al., 2023). We also test it in control tasks with both continuous states and actions, i.e. *MO-Hopper* (Basaklar et al., 2023; Alegre et al., 2023), *MO-Ant*, and *MO-Humanoid*. We assess the candidate methods using a variety of metrics sourced from Hayes et al. (2022).

#### 5.2.1 Benchmark environment setting

The deep sea treasure (DST) environment is a commonly used benchmark for the MORL setting (Vamplew et al., 2011; Abels et al., 2019; Yang et al., 2019; Alegre et al., 2023), see Figure 4(a). The agent needs to make a trade-off between the treasure value collected and the time penalty. It is featured by a discrete state space and action space. The state is the coordinate of the agent, i.e. [x,y]. The agent can move either *left*,

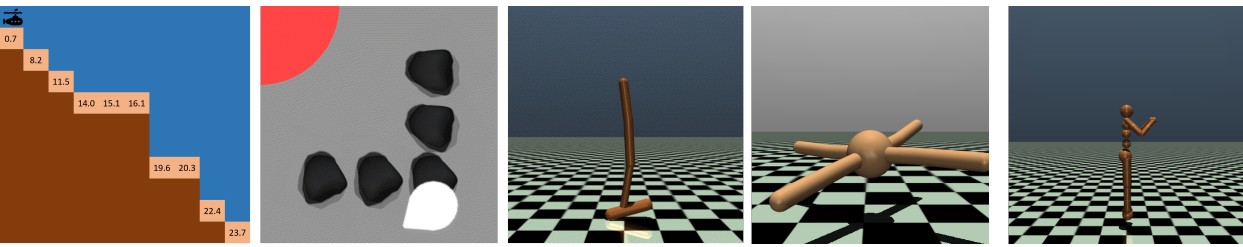

Figure 4: (a) Deep sea treasure; (b) Minecart; (c) MO-Hopper; (d) MO-Ant; (e) MO-Humanoid

*right*, *up*, *down* in the environment. The reward vector is defined as $\boldsymbol{r} = [r_{treasure}, r_{step}]$, where $r_{treasure}$ is the value of the treasure collected, and $r_{step} = -1$ is a time penalty per step. The treasure values are adapted to keep aligned with recent literature (Yang et al., 2019; Alegre et al., 2023). The discounted factor $\gamma = 0.99$.

*Minecart.* The Minecart environment (Abels et al., 2019; Alegre et al., 2023), see Figure 4(b), is featured with continuous state space and discrete action space. It provides a relatively sparse reward signal. The agent needs to balance among three objectives, i.e. 1) to collect ore 1; 2) to collect ore 2; 3) to save fuel. The minecart agent kicks off from the left upper corner, i.e. the base, and goes to any of the five mines consisting of two different kinds of ores and does collection, then it returns to the base to sell it and gets delayed rewards according to the volume of the two kinds of ores.

This environment has a 7-dimension continuous state space, including the minecart's coordinates, the speed, its angle cosine value and sine value, and the current collection of the two kinds of ores (normalized by the capacity of the minecart). The agent can select actions from the 6-dimension set, i.e. *mine*, *turn left*, *turn right*, *accelerate*, *brake*, *idle*. The reward signal is a 3-dimensional vector consisting of the amount of the two kinds of ore sold and the amount of fuel it has used during the episode. The reward for selling ores is sparse and delayed while the reward signal of fuel consumption is a constant penalty plus extra consumption when doing acceleration or mining.

MO-Hopper, MO-Ant and MO-Humanoid are extensions of the Mujoco continuous control tasks from OpenAI Gym (Borsa et al., 2019). They are used in literature to evaluate MORL algorithm performance in robot control (Basaklar et al., 2023; Alegre et al., 2023).

*MO-Hopper.* The agent controls the three torque of a single-legged robot to learn to hop. This environment involves two objectives, i.e. maximizing the forward speed and maximizing the jumping height. It has a continuous 11-dimensional state space denoting the positional status of the hopper's body components. This environment has a 3-dimensional continuous action space, i.e. the different torques applied on the three joints. We use the same reward function and discounted factor $\gamma = 0.99$ (Alegre et al., 2023).

$$r_{velocity} = v_x + C \tag{11}$$

$$r_{height} = 10(h - h_{init}) + C \tag{12}$$

where $v_x$ is the forward speed, $h$ and $h_{init}$ are the hoppers relative jumping height, $C = 1 - \sum_i a_i^2$ is the a feedback of the agent healthiness.

*MO-Ant.* The ant robot is a 3D model comprising a freely rotating torso with four legs attached. Each leg consists of two segments, making a total of eight body parts for the legs. The objective is to synchronize the movement of all four legs to propel the robot forward (toward the right) by exerting torques on the eight hinges that connect each leg's two segments to the torso. This configuration results in nine body parts and eight hinges in total that need to be coordinated for movement. This environment has an 8-dimensional continuous action space and 27-dimensional continuous state space (Schulman et al., 2015). We use the discounted factor $\gamma = 0.99$.

$$r_x = v_x + C \tag{13}$$

$$r_y = v_y + C \tag{14}$$

Table 1: Benchmark Environments

| Environment | DST | Minecart | MO-Hopper | MO-Ant | MO-Humanoid |
|---|---|---|---|---|---|
| State space $\mathcal{S}$ | discrete | continuous | continuous | continuous | continuous |
| $dim(\mathcal{S})$ | 2 | 7 | 11 | 27 | 376 |
| Action space $\mathcal{A}$ | discrete | discrete | continuous | continuous | continuous |
| $dim(\mathcal{A})$ | 4 | 6 | 3 | 8 | 17 |
| $dim(\mathcal{R})$[1] | 2 | 3 | 2 | 2 | 2 |
| Episode max horizon | 100 | 30 | 500 | 500 | 500 |
| Discounted factor[2] | 0.99 | 0.98 | 0.99 | 0.99 | 0.99 |
| Passing Percentage | 1 | 1 | 0.8→0.99 | 0.6→0.99 | 0.6→0.99 |
| Rollback Span[3] | 2 | 2 | 100 | 100 | 100 |

[1] The number of objectives / the dimension of reward vectors.

[2] Consistent with baselines from Alegre et al. (2023).

[3] The number of steps rolled back when the found trajectory surpasses the given demonstration.

where $v_x$ and $v_y$ is the x-velocity and the y-velocity, $h$ and $h_{init}$ are the hoppers relative jumping height, $C$ is the a feedback of the agent healthiness.

*MO-Humanoid.* The 3D bipedal robot is engineered to mimic human movement, featuring a torso (abdomen) equipped with two legs and two arms. Each leg is composed of three segments, simulating the upper leg, lower leg, and foot, while each arm consists of two segments, analogous to the upper and lower arms. The primary objective of this simulation environment is to enable the robot to walk forward as swiftly as possible while maintaining balance and avoiding falls. This environment has a 17-dimensional continuous action space and 376-dimensional continuous state space (Tassa et al., 2012). We use the discounted factor $\gamma = 0.99$.

$$r_{forward} = v_x + C \tag{15}$$

$$r_{control} = -10 * \sum control \tag{16}$$

where $v_x$ is the forward speed, $\sum control$ is the sum control cost of the joints $C$ is the a feedback of the agent healthiness.

To more effectively compare the exploration efficiency of our algorithm with baseline methods, we impose constraints on episode lengths across different environments. Specifically, we set the DST environment to terminate after 100 steps, the Minecart environment to conclude after 30 steps, and the MO-Hopper, MO-Ant and MO-Humanoid to end after 500 steps. This limitation intensifies the difficulty of the tasks, compelling the agent to extract sufficient information from a restricted number of interactions. This constraint ensures a more rigorous assessment of each algorithm's ability to efficiently navigate and learn within these more challenging conditions. The summary of benchmark settings is illustrated in Table 1.

### 5.2.2 Metrics

To thoroughly compare our algorithms with baselines, we use multiple metrics (Hayes et al., 2022).

*Expected utility (EU)* (Zintgraf et al., 2015): Compared to many other metrics, such as the hypervolume metric, EU is better suited for comparing different algorithms because they are specifically designed to evaluate an agent's ability to maximize user utility, which is consistently our primary objective. It is calculated as $EU(\Pi) = E_{\mathbf{w} \sim \mathcal{W}}[\max_{\pi \in \Pi} v_{\mathbf{w}}^{\pi}]$. The expectation in this work is the average over 100 evenly distributed weights from $\mathcal{W}$.

*Hypervolume* (Zitzler & Thiele, 1999): The Hypervolume metric quantifies the volume in the value-space that is Pareto-dominated by a set of policies within an approximate coverage set. We use reference point [0,0] in all benchmark environments except [-1,-1,-2] in Minecart.

*Sparsity* : To enhance the evaluation of Pareto front approximations, Xu et al. (2020) introduced a method that integrates the hypervolume metric with a sparsity metric. This approach aims to assess solution sets not only for their expansive coverage of the Pareto front but also for their distribution and separation, providing users with diverse and well-distributed options across the value space. The sparsity is defined as: $Sparsity(S) = \frac{1}{|S|-1} \sum_{j=1}^{m} \sum_{i=1}^{|S|} (\widetilde{S}_j(i) - \widetilde{S}_j(i+1))^2$, where $S$ is the PF approximation, $m$ is the number of objectives, $\widetilde{S}_j(i)$ is the $i$-th element of the sorted solutions on $j$-th objectives in $S$.

*Hypervolume/Sparsity*: As Pareto front approximations with a higher hypervolume metric while maintaining a lower sparsity metric are deemed better (Hayes et al., 2022), we assess the ratio of Hypervolume and Sparsity as another metric.

*PF Approximation*: We also present a visualization of the PF approximation to offer a clear understanding of the policy set that has been learned.

### 5.3   Result

In this section, we show the results of the evaluations of EU, PF, Hypervolume, Sparsity, and the ratio of Hypervolume and Sparsity. We conduct 5 independent runs with randomly picked seeds. For the initial number of demonstrations, we used 10 for DST, 7 for Minecart, and 5 for MO-Hopper, MO-Ant, and MO-Humanoid. We have conducted a sensitivity study of different initial demonstration quantities is in Appendix A.1. The sensitivity study of qualities of the guidance policy is in Appendix A.2. The experiments are run on a machine with 12th Gen Intel(R) Core(TM) i9-12900 CPU, NVIDIA T1000 graphic card and 32 GB memory. For the number of initial demonstrations, we use 10 demonstrations for DST, 7 demonstrations for Minecart, and 5 demonstrations for MO-Hopper, MO-Ant and MO-Humanoid.

### 5.3.1   Main experiment result

The evaluation of the EU of the training process is shown in Figure 5 (the evaluation frequency is shown in the title of each diagram). Demonstrations from different sources are used to train DG-MORL, i.e. hard-coded action sequences in DST and Minecart, and the underdeveloped policy of a TD3 agent in MO-Hopper, MO-Ant, and MO-Humanoid. This showcases our hypothesis that DG-MORL can work with demonstrations from any source. All evaluations of DG-MORL are solely with its *exploration policy*. To statistically and reliably evaluate the results of EU, we use stratified bootstrap confidence intervals (with 50000 runs) and interquartile mean (IQM) (Agarwal et al., 2021), i.e. IQMs are depicted with solid lines, while 95% confidence intervals are illustrated by color-matched shaded areas.

In Figure 5, the result shows that DG-MORL has learned a better or at least competitive policy than the baselines in the first evaluation in most benchmarks. An exception occurs in MO-Ant, where DG-MORL starts with an EU approaching 0. This may be due to the more conflicting objectives in MO-Ant, where initial demonstrations display contradictory behaviors, e.g. exclusively moving along the x-axis or y-axis. Starting from a demonstration committed to a single direction can impede progress in improving the policy going in the alternative direction. Conversely, DG-MORL in MO-Humanoid exhibits a notably strong start, attributed to its objectives of forward movement and control cost reduction, which approximately align as a single goal of advancing with minimal control energy (similar to MO-Hopper). This alignment results in less contradictory demonstrations than MO-Ant, as further evidenced by the sparse points on the PF illustrated in Figure 6. This phenomenon indicates that the degree of conflict among objectives can be a potential factor influencing the training process of DG-MORL. However, after training DG-MORL achieves better performance than all of the baselines, surpassing the initial demonstrations and reaching the standard of the best achievable performance[5]. In comparison, GPI-PD also attains a commendable level of performance in the early stages, but during the whole training process, it shows larger variations and inferior

---

[5]For DST and Minecart, where a known PF exists, the black dashed line indicating the best performance represents the results from the actual PF. As MO-Hopper, MO-Ant, and MO-Humanoid do not have known PFs, we trained 5 single-objective TD3 (SO-TD3) agents with preferences [1, 0], [0.75, 0.25], [0.5, 0.5], [0.25, 0.75], [0,1]. Each agent is trained for 1.5M steps, therefore the training budget is 7.5M steps. We use the result as the best performance (Note that these "best" performances can be exceeded as they only from policy trained with significantly more steps). Our algorithm is trained by 0.225M steps, which takes only 3% training budget compared to SO-TD3.

final performance than DG-MORL. The GPI-LS algorithm exhibits significantly greater variances in all environments than DG-MORL suggesting reduced robustness. It is also noted that GPI-PD's superiority over GPI-LS diminishes and is eventually surpassed by GPI-LS in the MO-Humanoid environment, highlighting the challenges model-based MORL faces in complex tasks.

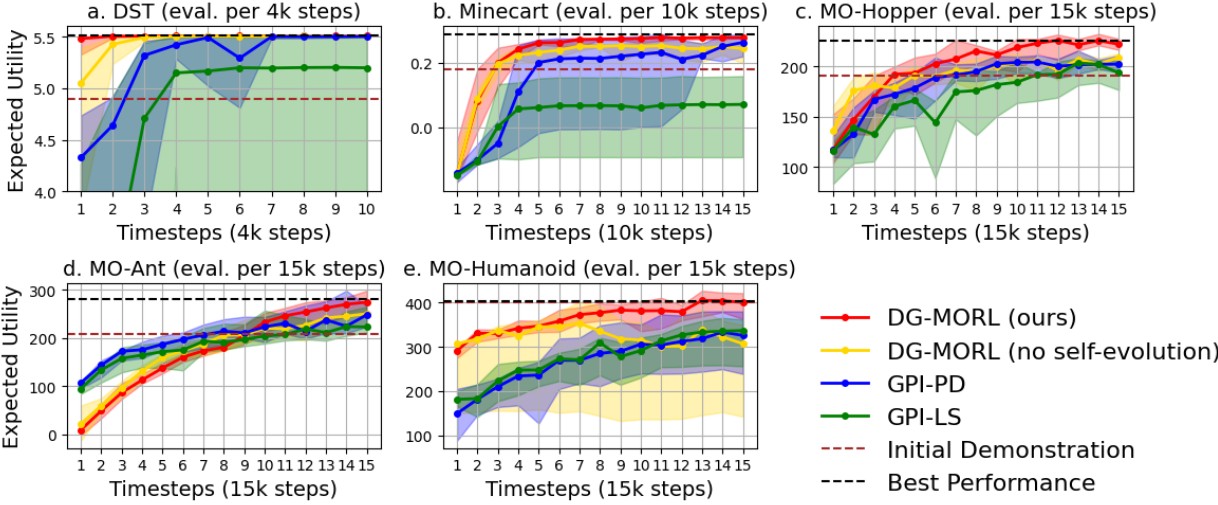

Figure 5: EU evaluation (a) DST; (b) Minecart; (c) MO-Hopper; (d) MO-Ant; (e) MO-Humanoid. The numbers in brackets on the x-axis represent the interval of timesteps between each evaluation. The result shows that DG-MORL has learned a better or at least competitive policy in most environments.

In the DST, DG-MORL successfully reconstructs the ground truth PF. We therefore did not conduct the PF visualization and Hypervolume, Sparsity evaluation for DST. For Minecart, where the PF is 3-dimensional and challenging to interpret visually and the environment is stochastically initialized, we have confined representation to only EU metric. Figure 6 illustrates the PF approximation in the other three environments. In MO-Hopper and MO-Ant, DG-MORL's PF dominates both baseline approaches. Although in MO-Humanoid, DG-MORL's PF does not dominate that of GPI-PD, it does dominate GPI-LS. The PF approximation results for MO-Humanoid reveal an interesting observation: the solutions for each method tend to cluster, suggesting that despite being a multi-objective task, MO-Humanoid behaves similarly to a single-objective task. This observation underscores the need for more thoughtfully designed MORL benchmark environments that can more effectively differentiate the performance of various algorithms. Furthermore, as indicated by Figure 5 and Table 2, DG-MORL demonstrates superior EU, higher Hypervolume, lower Sparsity, and improved Hypervolume/Sparsity ratio compared to GPI-PD, suggesting a significantly enhanced quality of the solution set. The PFs reveal that DG-MORL surpasses the initial demonstrations in all three environments, while one instance of the best demonstrations in MO-Ant outperforms some solutions of DG-MORL.

Table 2 shows the results for Hypervolume and Sparsity [6]. DG-MORL records the highest Hypervolume across most environments. Although DG-MORL's Sparsity and Hypervolume/Sparsity ratios do not consistently achieve top marks, they still maintain competitive levels. In instances where DG-MORL's Sparsity and Hypervolume/Sparsity are exceeded, GPI-LS achieves the best results. Notably, DG-MORL (no self-evolving) sees a Sparsity of 0 in MO-Humanoid, attributable to the single point on its PF, which limits interpretative value. Nonetheless, while GPI-LS has higher Sparsity, this comes at the cost of overall performance in terms of EU, Hypervolume, and PF coverage.

---

[6]The $\uparrow/\downarrow$ means the higher/lower the value, the better the performance.

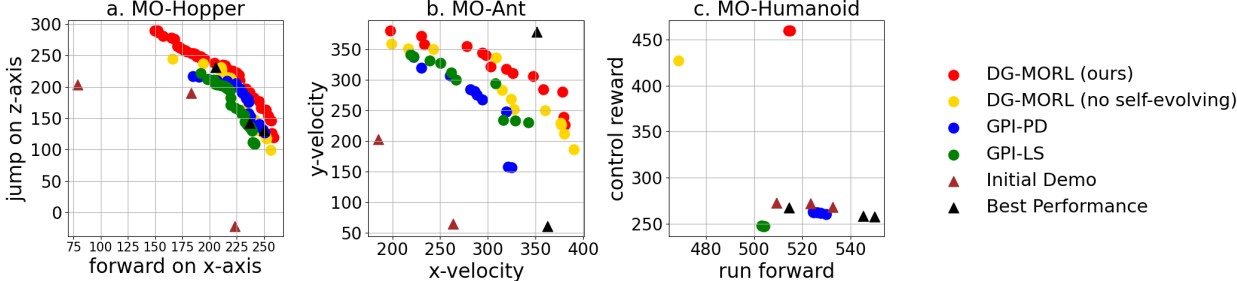

Figure 6: PF approx. (a) MO-Hopper; (b) MO-Ant; (c) MO-Humanoid; In MO-Hopper and MO-Ant, DG-MORL's PF dominates both baseline approaches. Although in MO-Humanoid, DG-MORL's PF does not dominate that of GPI-PD, it does dominate GPI-LS.

Table 2: Evaluations on Hypervolume and Sparsity

| Method | Hypervolume↑ | Sparsity↓ | Hypervolume/Sparsity↑ |
|---|---|---|---|
| MO-Hopper | | | |
| DG-MORL | **68343.44** | 36.95 | 1849.49 |
| GPI-PD | 52710.74 | 57.92 | 910.03 |
| GPI-LS | 51545.64 | **24.87** | **2072.96** |
| DG-MORL (no self-evolving) | 58508.60 | 163.55 | 357.74 |
| Initial Demo | 35915.02 | 29010.59 | 1.24 |
| Best Performance | 53766.39 | 4603.11 | 11.68 |
| MO-Ant | | | |
| DG-MORL | **136021.51** | **607.24** | **224.0** |
| GPI-PD | 103513.67 | 1292.10 | 80.11 |
| GPI-LS | 110327.95 | 762.32 | 144.73 |
| DG-MORL (no self-evolving) | 128635.06 | 1073.22 | 119.86 |
| Initial Demo | 42682.51 | 25469.52 | 1.68 |
| Best Performance | 133711.36 | 101590.39 | 1.32 |
| MO-Humanoid | | | |
| DG-MORL | **236740.17** | 0.43 | **545903.71** |
| GPI-PD | 139359.20 | 2.80 | 49784.78 |
| GPI-LS | 125137.57 | **0.28** | 451889.30 |
| DG-MORL (no self-evolving) | 200264.31 | 0.00 | N/A |
| Initial Demo | 145364.09 | 149.35 | 973.28 |
| Best Performance | 146949.70 | 523.19 | 280.87 |

### 5.3.2 Ablation study

We conduct an ablation study on DG-MORL by excluding the *self-evolving mechanism*. The agent relies solely on the initial demonstrations, as opposed to progressively enhancing the guidance demonstration with self-generated data. The result is shown in yellow in Figure 5, Figure 6, and DG-MORL (no self-evolving) in Table 2. The absence of this mechanism makes DG-MORL fail to reach the performance levels of the fully equipped DG-MORL setup (except in DST), however, it still outperform the baselines and the initial demonstrations in most benchmarks (except the MO-Humanoid, the instance without self-evolving saw a drop in the middle.). The findings from the ablation study confirm the following: 1. The *self-evolving mechanism* is effective. 2. Generally, using demonstrations to guide MORL training leads to improved performance.

# 6 Limitation

DG-MORL has pioneered the use of demonstrations to inform MORL training. However, our method has some limitations. One limitation of this work is that DG-MORL is only applicable to linear preference weights. It needs further improvement to work with non-linear preferences. Another limitation is that DG-MORL has not been evaluated in a non-stationary environment. It is worthwhile to explore how a demonstration can guide the training process in non-stationary MORL settings. We will leave these to the future. Additionally, since DG-MORL in the MO-Ant scenario does not achieve jump-start learning, exploring how to effectively use demonstrations to aid training in environments with relatively contradictory objectives is worth further investigation.

# 7 Conclusion

We proposed the pioneering DG-MORL algorithm that uses demonstrations to enhance MORL training efficiency. Our algorithm has overcome the challenge of sparse reward, the hard beginning of training, and the possible derailment from a complex environment. We address the challenges of demonstration-preference misalignment and demonstration deadlock by implementing corner weight computation of the demonstrations. We introduced the *self-evolving mechanism* to handle sub-optimal demonstrations or the case with limited number of demonstrations. Empirical evidence confirms DG-MORL's superiority over state-of-the-art across various environments. Additionally, our experiments highlight DG-MORL's robustness by excluding self-evolving and under other unfavoured conditions. We also provided theoretical insights into the lower bound and upper bound of DG-MORL's sample efficiency.

**Acknowledgments**

This work was financially supported by the Irish Research Council under Project ID: GOIPG/2022/2140. The authors have applied a CC BY 4.0 public copyright license to the Author Accepted Manuscript version arising from this submission to ensure Open Access. We would also like to express our sincere appreciation to the reviewers for their insightful and constructive feedback, which has played a crucial role in refining our research and strengthening the final manuscript.

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

# A  Sensitivity study

It is widely acknowledged in machine learning that both the quantity and quality of data play crucial roles in enhancing the outcomes. To understand the extent and manner in which each of these factors impacts the learning process, we have carried out a series of sensitivity studies. We continue to utilize the same five benchmark environments and the EU metric. However, the focus shifts to a comparison among DG-MORL agents rather than among other baseline algorithms.

In the first part, we explore the sensitivity of the DG-MORL algorithm to the quantity of initial demonstrations. The second part of this section examines how the quality of data impacts the algorithm's performance. For better comparison, the GPI-PD result is included for reference depicted by a blue dashed line.

## A.1  Sensitivity to initial demonstration quantity

Figure 7, 8, 9, 10, and 11 shows DG-MORL algorithm's sensitivity to the quantity of initial demonstrations in the three benchmark environments separately. The figures presented depict the average results of 5 independent runs for the sake of clarity. It should be noted that scenarios without any initial demonstrations are not taken into account, as the presence of such demonstrations is a fundamental prerequisite for DG-MORL. We ensure that the process commences with a minimum number of demonstrations equivalent to the number of objectives, e.g. for DST, as there are 2 objectives, the least amount of demonstration provided should be 2 as well.

Though over time, there are more demonstrations added by the *self-evolving mechanism*, the outcomes of our research reveal a clear relationship between the quantity of initial demonstrations and the performance of the DG-MORL algorithm, particularly evident in the DST environment. Specifically, there is a positive correlation observed: as the number of initial demonstrations increases, there is a notable enhancement in training efficiency and a corresponding improvement in policy performance.

This trend is consistent in the Minecart environment as well. Here too, the quantity of initial demonstrations plays a significant role in influencing the performance of the DG-MORL algorithm. The increased number of initial demonstrations provides more comprehensive guidance and information, which in turn facilitates more effective learning and decision-making by the algorithm. This consistency across different environments underscores the importance of initial demonstration quantity as a key factor in the effectiveness of the DG-MORL algorithm.

An intriguing observation from the Minecart experiment was that an increase in the number of initial demonstrations actually led to lower performance at the beginning of training. This phenomenon was not evident in the DST experiment, suggesting a unique interaction in the Minecart environment. We hypothesize that this initial decrease in performance is attributable to the complexity of the Minecart environment when paired with a large set of demonstrations. In such a scenario, the agent may struggle to simultaneously focus on multiple tasks or objectives presented by these demonstrations, leading to a temporary dip in performance at the early stages of training. This is indicative of an initial overload or confusion state for the agent, as it tries to assimilate and prioritize the extensive information provided by the numerous demonstrations. However, as training progresses and the agent becomes more familiar with the environment and its various tasks, it starts to overcome this initial challenge. With time, the agent effectively integrates the insights from the demonstrations, leading to improved performance. This dynamic suggests that while a wealth of demonstrations can initially seem overwhelming in complex environments, they eventually contribute to the agent's deeper understanding and better performance, underscoring the importance of considering both the quantity and nature of demonstrations in relation to the environment's complexity.

The results from the MO-Hopper, MO-Ant further support the notion that a greater number of initial demonstrations tends to yield better performance. However, in MO-Humanoid, the quantity of initial demonstrations have no significant influence on the learning process. This may be because that MO-Humanoid environment from the reward side is more like a single objective problem and any reasonable demonstration can provide adequate guidance.

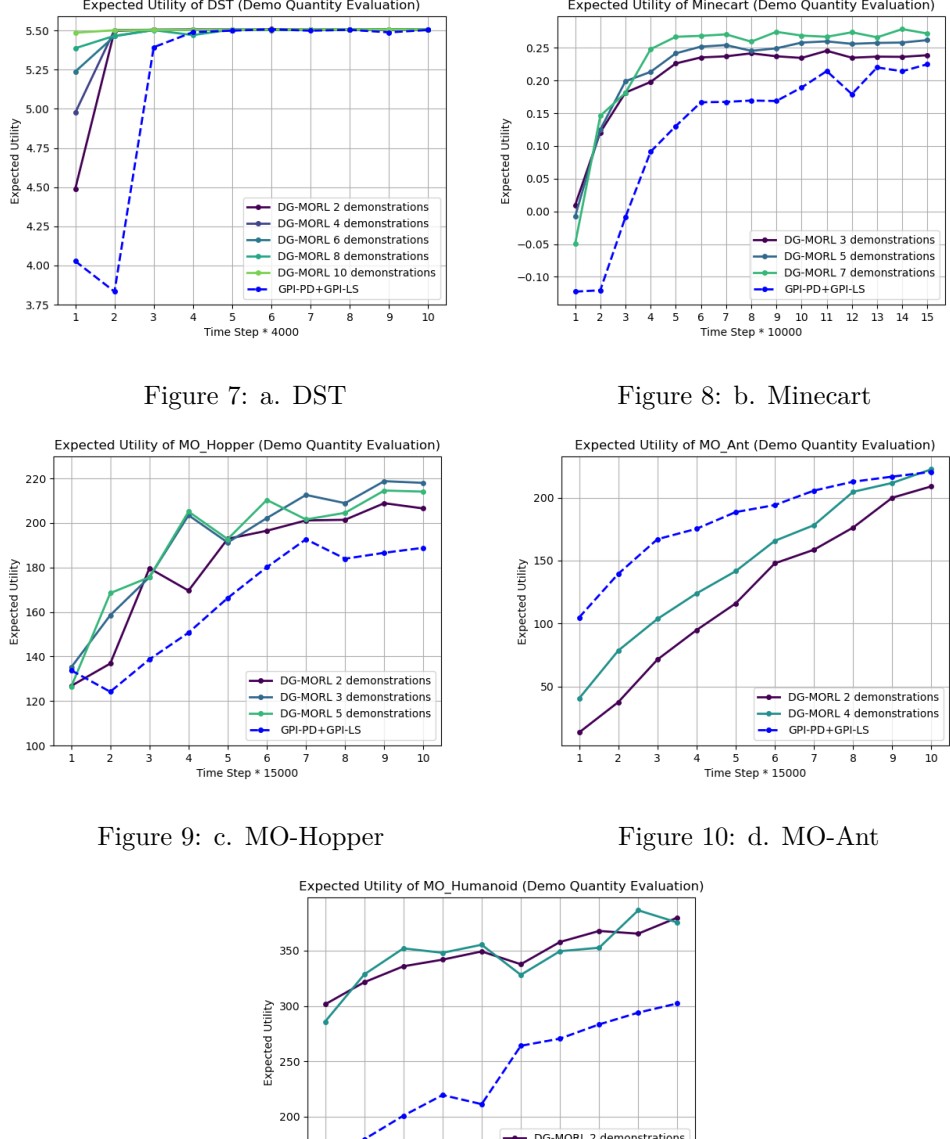

Figure 7: a. DST

Figure 8: b. Minecart

Figure 9: c. MO-Hopper

Figure 10: d. MO-Ant

Figure 11: e. MO-Humanoid

However, an interesting deviation was observed in MO-Hopper and MO-Humanoid where the performance with fewer initial demonstrations was marginally better than with more. This outcome implies that, beyond a certain point, additional demonstrations might not always contribute positively to the learning process. Potential reasons for this include:

Conflicting Action Selection: More demonstrations could introduce complexity in terms of conflicting action choices in similar states under different preferences. This conflict might hinder the agent's ability to learn a coherent and effective policy. This is further emphasized by the result of the MO-Ant, as the nature of the two objectives is very conflicting. The learning process is therefore falling behind the GPI-PD curve at the start, though the one with more demonstrations catches up the performance of GPI-PD at last.

Complexity and Long Horizon: The intrinsic complexity of the MO-Hopper, MO-Ant and MO-Humanoid, along with their extended horizon, might complicate the learning process. When the agent learns an effective policy that surpasses some of the *guide policies*, it might inadvertently "forget" how to surpass others, particularly those aligned with different preferences.

These observations indicate that while having a sufficient number of demonstrations is beneficial, there is a nuanced balance to be struck. Too many demonstrations, especially in complex environments, can introduce new challenges that potentially offset the benefits of additional information.

Remarkably, the DG-MORL algorithm demonstrates its robust learning capabilities by outperforming the GPI-PD baseline (except in MO-Ant, it needs more training steps to surpass GPI-PD, see Section 5.3.1. However, it can catch up the performance of GPI-PD at last.), even when provided with a limited number of initial demonstrations. This aspect of DG-MORL underscores its efficacy in leveraging available resources to achieve superior learning outcomes, highlighting its potential for applications where abundant demonstration data may not be readily available.

## A.2  Sensitivity to initial demonstration quality

The quality of initial demonstrations emerges as another factor influencing the training process. While the *self-evolving mechanism* can to some degree mitigate this impact, it is still presumed that initial demonstrations of higher quality are associated with more effective learning outcomes because they may lead the agent to more potential areas in the state space.

Figures 12, 13, 14, 15, and 16 collectively illustrate the DG-MORL algorithm's responsiveness to the quality of initial demonstrations across three different environments. These figures compare the overall performance of the DG-MORL agent when initialized with varying levels of demonstration quality. Additionally, they present the EU for each set of demonstrations, indicated by dashed lines in colors corresponding to each demonstration standard.

In the DST environment, for instance, the learning outcomes for all three categories of initial demonstrations (high, medium, and low quality) eventually reach the environment's upper-performance limit. However, noteworthy differences are evident in the initial phases of training. The DG-MORL agent initialized with the highest quality demonstrations achieves significantly better performance right from the first evaluation round. The performance curve of the agent with medium-quality demonstrations shows a slightly quicker ascent compared to the one with lower-quality demonstrations. This pattern underscores the impact of initial demonstration quality on both the speed and efficiency of learning, especially in the early stages of training.

In the Minecart, MO-Hopper, MO-Ant and MO-Humanoid environments, the performance outcomes for the three sets of initial demonstrations reveal a notable observation: the quality of the demonstrations has a relatively minor influence on the final learning results. This outcome contrasts with the more pronounced effect seen in the DST environment. In Minecart, MO-Hopper, MO-Ant and MO-Humanoid, despite variations in the quality of the initial demonstrations (high, medium, and low), the differences in the final learning outcomes are less significant (but still higher). This suggests that, in these particular environments, the DG-MORL algorithm is capable of achieving comparable levels of performance regardless of the initial quality of demonstrations.

This could be attributed to the inherent characteristics of the environments, where the algorithm might have more flexibility or capability to compensate for the initial quality of demonstrations during the learning process. It highlights the adaptive nature of the DG-MORL algorithm and suggests that its performance is not solely dependent on the quality of initial demonstrations, especially in certain types of environments.

Similar to the observations made in the experiment evaluating the impact of demonstration quantity, the DG-MORL algorithm consistently outperformed the GPI-PD method in all scenarios (except in MO-Ant, where DG-MORL only caught up with the performance of GPI-PD at last. But it does have outperformed GPI-PD in longer training steps, see Section 5.3.1, therefore does not harm the superiority of DG-MORL) tested for demonstration quality. This consistent superiority across different conditions and environments empirically validates the efficacy and robustness of the DG-MORL algorithm. It demonstrates that DG-MORL can effectively utilize the information provided in the demonstrations, whether of high or lower quality, to

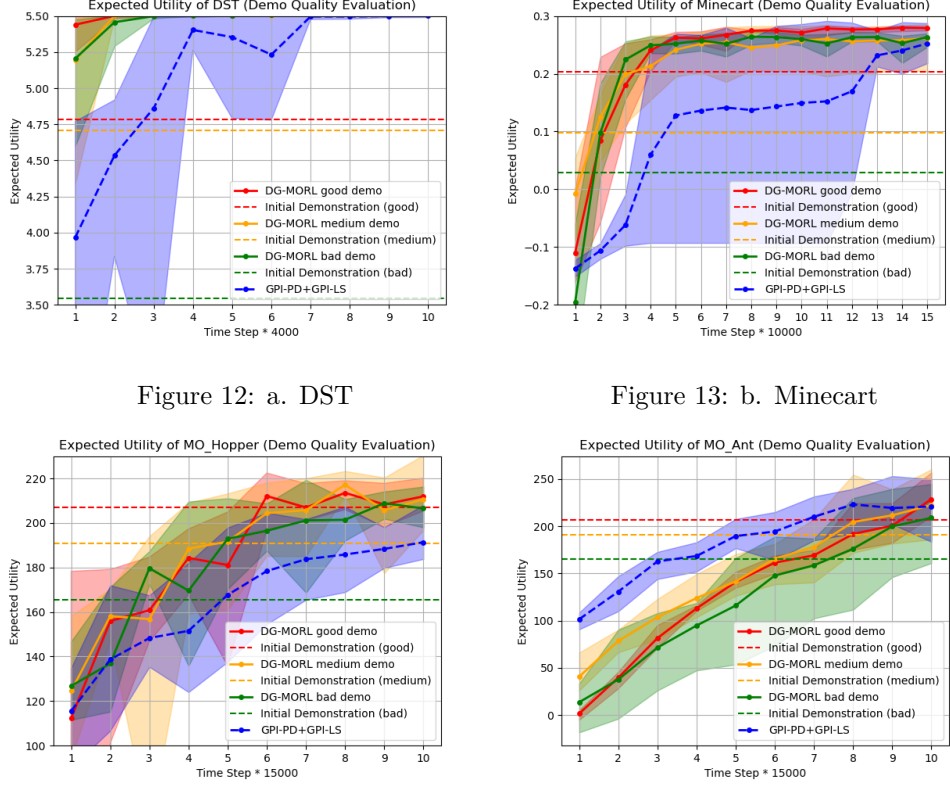

Figure 12: a. DST

Figure 13: b. Minecart

Figure 14: c. MO-Hopper

Figure 15: d. MO-Ant

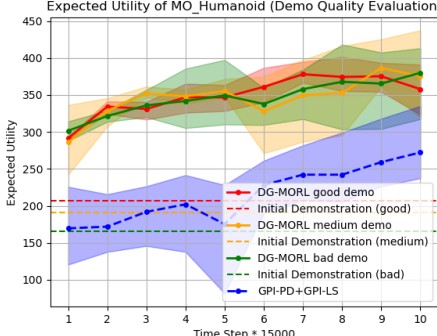

Figure 16: e. MO-Humanoid

enhance its learning process and achieve superior outcomes. This finding is significant as it not only attests to the performance capabilities of DG-MORL but also reinforces its potential applicability in a wide range of real-world scenarios where the quality of available demonstrations may vary.

