# OpenReview forum: "Demonstration-Guided Multi-Objective Reinforcement Learning"
_TMLR — Accepted by TMLR_

### Review · Reviewer_kgw3 · 2024-10-02

**Summary Of Contributions:**

The paper introduces DG-MORL, a multi-objective reinforcement learning method leveraging prior demonstrations. It outlines and addresses several common issues with existing MORL methods and also proposes a self-evolving mechanism to refine the demonstration set, preventing sub-optimal examples from hindering progress. It also provides some theoretical results on sample complexity and benchmarks its performances against known other MORL algorithms.

**Audience:**

Yes

**Claims And Evidence:**

Yes

**Requested Changes:**

See weaknesses section. A few more details:
- Section 4.1: intuitive explanation of corner weights not very clear
- Section 4.2: How do you derive the demonstration policies in the case where the demonstrations do not cover all states?
- In general section 4 is very unclear and some practical questions (such as the examples above) are not addressed enough
- I would suggest adding much more explanation on the construction of DG-MORL and the self-evolving mechanism to make the derivation of the full model more intuitive

**Strengths And Weaknesses:**

Strengths:
- DG-MORL is an interesting approach - integrating information from prior demonstrations seems to be a fairly natural idea and it is good to have it analyzed in a rigorous setting
- Paper is well structure and clearly outlines the different problems in MORL that are addressed
- Theoretical analysis gives a good mathematical grounding to the method

Weaknesses:
- Some sections are unclear and are hard to follow - particularly section 4
- I would have expected a more thorough literature review, e.g. encompassing Bayesian methods which can be an naive way of integrating the information contained in the demonstrations
- Very little details on the construction of demonstrations and the case of suboptimal demonstrations is not treated at all in the experiments - while proving the value of adding demonstrations to MORL is certainly good, I believe that a deeper analysis of the added value of suboptimal demonstrations would significantly increase the novelty of the paper's contribution

---

> ### Author Response · Authors · 2024-10-08
> **We appreciate your hard work and the valuable feedback you've provided**
>
> Dear Reviewer kgw3,
>
> We sincerely appreciate the time and effort you have dedicated to reviewing our manuscript, as well as your valuable and constructive feedback. Your suggestions have been incredibly helpful, and we are grateful for your insights.
>
> We are currently in the process of revising the manuscript to address your concerns and implement the requested changes. A revised version will be uploaded shortly, along with a detailed rebuttal.
>
> Thank you once again for your thoughtful review and support.
>
> Best regards,
>
> The authors of manuscript 3264

---

### Review · Reviewer_SfsV · 2024-10-10

**Summary Of Contributions:**

Broadly, the paper studies the problem of using prior demonstrations for bootstrapping policies in multi-objective reinforcement learning (MORL). This problem of leveraging prior demonstrations is challenging in multi-objective RL, since the preference weights over the objectives under the which the demonstration was drawn are typically unknown. The authors build upon prior work on MORL and the Jump Start RL approach (Uchenda et al., 2023). Specifically, under the assumption of a linear utility (which would capture only the convex hull of the Pareto front), the authors propose an approach, called DG-MORL, which consists of maintaining a set of demonstrations as guide policies, inferring the corner weights from the vectorized returns of the demonstrations, mixing the guide policy with an exploratory policy to collect better demonstrations, following a cirriculum controlling the mixing fraction. The authors present some theoretical analysis (more on this below) on the sample complexity of the proposed approach. Finally the authors evaluate DG-MORL on four tasks and observe that empirically the approach outperforms the baselines.

**Audience:**

Yes

**Broader Impact Concerns:**

The authors do not discuss any broader impact, since their work is about general algorithmic improvements with no direct applications with broader impact.

**Claims And Evidence:**

No

**Requested Changes:**

* The theoretical results section needs a complete revision based on the Weaknesses section and the comments below.
* Experiments on one more domain with an unstructured observation space would make the results much more reliable.
* The authors should consider designing stronger baselines based on off-policy MORL algorithms (e.g. in the Weaknesses section above)
* Additionally, the authors claim one of the key contributions to be, what they call "self-evolving mechanism". I find it confusing, since what it really seems to be is just shifting to on-policy training as more experience is collected. I am not sure if it is helpful to introduce new terminology for something which can be described with the standard terminology.
* There are a lot of writing issues some of which I have listed below. Overall, a thorough proof read would improve the paper considerably.



Specific changes:
* Throughout the paper, the equations are not typeset correctly. Please use TeX commands when typesetting equations. e.g. in the entire paper the authors have used `max` instead of `\max` which makes the equations much harder to read.
* Throughout the paper, the wrong quotations are used. Please use ``...'' instead of "".
* Page 1, paragraph 2, L1: "magnifies" -> "amplifies" / "exacerbates"
* Page 2, paragraph 1, last line: "development of a sub-optimal policy" -> "learning of a sub-optimal policy".
* Page 3, paragraph 1, last line: what does "most potential preference" mean?
* Page 4, paragraph 5: CCS is used before being defined. (I assume it is convex coverage set?)
* Figure 1: The color scheme in the figure could be improved. Currently it is hard to distinguish between the orange and red circles.
* Page 5, above equation (1):  "proposed by (Roijers, 2016)" should be using `\citet` instead.
* Page 5, below Theorem 1, last sentence: "most proper corner weight". What does most proper mean here?
* Page 6, Section 4.4, L1: "DG-MORL pioneers" -> "DG-MORL introduces"
* Page 6, Section 4.4, L3: "commendable outcomes" -> "strong performance"
* Page 6, Section 4.4, paragraph 3, L5: The sentence "the better demonstration is popped into the demonstration repository to achieve self-evolving" is poorly phrased.
* Algorithm 1:
    * L7-L9: What does it mean for a variable e.g. $\mathbf{w}_c$ to be "=" to an equation? Something like "Set $\mathbf{w}_c$ based on Equation 2" is more appropriate.
    * L9: why is $h$ set equal to $H$? Shouldn't it be a hyperparameter? If they are set equal then there is no mixing.
    * L13: How is $\pi_e$ trained? There is no discussion of this anywhere in the paper.
* Page 7, Section 4.5 L1: "We provide a theoretical analysis of the algorithm sample efficiency’s lower bound and upper bound" -> "We provide a theoretical analysis of the lower bound and upper bound on the sample complexity of the algorithm"
* Page 9, Assumption 1: What do $d^{\pi^*}$ and $d^{\pi_g}$ mean? What does the constant $C$ denote?
* Page 9, Corollary 3: The authors write "We have provided the upper bound of the guide policy under a self-evolving mechanism". Upper bound on what quantity?
* Page 9, definition 2: What is $\mathcal X$? The reader should not have to open another paper to understand the notation for a definition in the the paper.

**Strengths And Weaknesses:**

**Strengths:**
* The problem the paper studies is quite interesting, practically relevant and understudied. To the best of my knowledge this is one of the first approach for MORL which can incorporate prior demonstrations with unknown preferences.
* The overall approach is relatively simple and straightforward, mostly extending the JSRL work to a multi-objective setting, which I view as a positive.
* While there are some issues with the experiments (discussed below), the results demonstrate the performance gains over the GPI baselines.

**Weaknesses:**
I think there are several places where the paper needs some major revisions.

* A primary area of concern for me is the section on the theoretical analysis. Specifically, the upper bound analysis. There are several issues in the analysis. First, there are several quantities used without being defined. A lot of the content is taken directly from Uchenda et al., (2023), but not explained in sufficient detail. The proof for the theorem is missing (even if it is taken from a prior work and adapted to the setting it would be good to have the proof). More importantly, the authors simply analyze the multi-objective contextual bandit setting and use that as an upper bound for their approach with the justification "our algorithm works based on
the Q-learning and TD3 paradigm [...] it
is supposed to be less likely to violate the assumption". This claim is incorrect and as such the analysis in the section, in my view is not meaningful. Some more issues are detailed in the requested changes section.

* Further, the authors only use a single baseline GPI-PD. The authors explain that this is because GPI outperforms other methods like Envelope-MOQ, PGMORL. While I can understand limitations due to compute for running every method, but I believe the baseline comparisons could be considered a bit more deeply. For example, while GPI-PD may perform better, an approach like Envelope-MOQ might be a better suited baseline since it is also an off-policy approach and in-principle a similar approach of computing corner weights can be used to have a stronger baseline.

* Additionally, the empirical validation seems somewhat limited. The authors test DG-MORL on four different tasks, however, they all appear to be relatively simple and small in scale (except the Humanoid task). I think other environments with high-dimensional observations (instead of the state itself), for example, the SuperMario task considered in Envelope-MOQ would strengthen the results.


* Overall, I think the writing quality is quite poor and the paper could have benefitted from a thorough proof-reading. There are several places where quantities are undefined, not introduced, details missing, and use of incorrect notation. (See some instances in the requested changes section).


Uchendu, Ikechukwu, et al. "Jump-start reinforcement learning." International Conference on Machine Learning. PMLR, 2023.

---

### Review · Reviewer_6uAU · 2024-10-14

**Summary Of Contributions:**

The authors tackle the problem of multi-objective RL. In this problem setting, we have multiple potentially-competing objectives. Rewards are vectors of multiple objectives rather than scalars, and our goal is to learn a Pareto frontier of non-dominated policies. The typical formulation for this is to have some preference weight vector $w$ as input to the policy.

Taking inspiration from methods like jump-start RL (Uchendu 2023), the goal is to speed up policy learning by using demonstrations to guide the policy learning. In JSRL, given some guide policy $\pi_g$, we run $\pi_g$ for $h$ steps, then run an exploratory policy for the remaining steps. As the exploratory policy's performance improves, $h$ is reduced to yield more control to exploratory policy.

The primary modification needed in this work is to support a multi-objective RL setup. The guide policies are a set of demonstrations, which (I believe) are simply replayed in the environment. Given a new candidate weight, we find the demonstration that best maximizes that candidate weight, and roll it out as the guide policy. As RL progresses, if new episodes exceed some fraction (passing percentage) of the return from the selected demonstration, that new episode is added as a new demonstration. This passing threshold starts low and increases over time, with the set of demonstrations updated over time to remove any dominated rollouts.

This method is compared to various generalized policy improvement (GPI) baselines, a standard technique for multi-objective RL.

**Audience:**

Yes

**Claims And Evidence:**

Yes

**Requested Changes:**

I would recommend include more text in figure captions (what is the important conclusion you are arguing from the figure?), and use fewer abbreviations.

**Strengths And Weaknesses:**

There are signs of better policy improvement compared to GPI baselines. The heavy use of acronyms makes it hard to keep track of what the paper is doing - IMO the clarity would be better if metrics like Hypervolume, Sparsity, etc were spelled out rather than abbreviated so readily.

Even after reading the paper more closely, it is still not clear to me how the guide policy is implemented. Is it a learned behavior cloning model, or simply a rollout of the existing demonstration? If it's the latter, the method seems very similar to Go Explore (which is cited in related work).

I overall think the paper has issues on clarity, but believe the results are reasonable.

---

### Decision · Action_Editor_Zvzk · 2024-11-13

**Recommendation:** Accept with minor revision

**Comment:**

Following the initial reviews of the paper, the authors have made several changes and corrections which seem to address all the concerns raised by the reviewers. There is a consensus that the paper now provides sufficient evidence to support its claims, and that this work synthesizes several important results relevant to TMLR's audience. I therefore recommend acceptance of the paper.

I note two minor points in the manuscript that the authors should address in the final version:

 - p.1 - (i) We propose the first MORL algorithm (DG-MORL) can use demonstrations to enhance the training. -> There is a typo

 - p.9 Fig. 3 - The text font in this figure is larger that the text in the manuscript. I recommend down-scaling that figure a little, and expand the figure's caption to be more explicit and self-contained (as recommended by reviewer 6uAU)

**Audience:**

All reviewers and myself agree that MORL and Decision-Guided RL (DG-RL) are relevant topics for TMLR and that this paper will find an audience.

**Claims And Evidence:**

This paper claims the following:
 - the first Multi-Objective Reinforcement Learning (MORL) algorithm that can use demonstrations to enhance training
 - a new self-evolving mechanism to adapt and improve the quality of guiding data during training

The authors present a literature review, a theoretical analysis of their proposed algorithm, as well as experimental results that compare their algorithms to baseline methods. Despite initial flaws in the original submission, all reviewers seem happy with the corrections in the new revision submitted by the authors, and acknowledge that it provides enough evidence to support the authors' claims.

---

> ### Author Response · Authors · 2024-11-15
> **Revision of the minor points and sincerely thank the great help from reviewers and AE.**
>
> Dear Action Editor Zvzk
> - We have revised " (i) We propose the first MORL algorithm (DG-MORL) can use demonstrations to enhance the training." as "(i) We propose the first MORL algorithm (DG-MORL) that can leverage demonstrations to improve training performance."
> - We have down-scaling Figure 3 to make it more explicit and self-contained with the caption.
> __________________________________________________________
> We sincerely thank Action Editor Zvzk, Reviewer 6uAU, Reviewer SfsV, and Reviewer kgw3 for their time and effort in reviewing and providing valuable feedback on our work. Your insightful comments and advice have significantly enhanced the quality and clarity of our research, making it more robust and concrete.
>
> Once again, we deeply appreciate your constructive guidance and support.
>
> Best regards,
> Authors of Manuscript 3264